# Hemocyanins from *Helix* and *Rapana* Snails Exhibit in Vitro Antitumor Effects in Human Colorectal Adenocarcinoma

**DOI:** 10.3390/biomedicines8070194

**Published:** 2020-07-05

**Authors:** Ani Georgieva, Katerina Todorova, Ivan Iliev, Valeriya Dilcheva, Ivelin Vladov, Svetlozara Petkova, Reneta Toshkova, Lyudmila Velkova, Aleksandar Dolashki, Pavlina Dolashka

**Affiliations:** 1Institute of Experimental Morphology, Pathology and Anthropology with Museum, Bulgarian Academy of Sciences, Sofia 1113, Bulgaria; pda54@abv.bg (K.T.); taparsky@abv.bg (I.I.); val_dilcheva@yahoo.com (V.D.); iepparazit@yahoo.com (I.V.); svetlozarapetkova@abv.bg (S.P.); reneta.toshkova@gmail.com (R.T.); 2Institute of Organic Chemistry with Centre of Phytochemistry, Bulgarian Academy of Sciences, Sofia 1113, Bulgaria; lyudmila_velkova@abv.bg (L.V.); adolashki@yahoo.com (A.D.)

**Keywords:** hemocyanin, snail *Rapana venosa*, snail *Helix lucorum*, snail *Helix aspersa*, antitumor activity, colorectal adenocarcinoma, apoptosis

## Abstract

Hemocyanins are oxygen-transporting glycoproteins in the hemolymph of arthropods and mollusks that attract scientific interest with their diverse biological activities and potential applications in pharmacy and medicine. The aim of the present study was to assess the in vitro antitumor activity of hemocyanins isolated from marine snail *Rapana venosa* (RvH) and garden snails *Helix lucorum* (HlH) and *Helix aspersa* (HaH), as well the mucus of *H. aspersa* snails, in the HT-29 human colorectal carcinoma cell line. The effects of the hemocyanins on the cell viability and proliferation were analyzed by 3-(4,5-dimethylthiazol-2-yl)-2,5-diphenyltetrazolium bromide (MTT) assay and the alterations in the tumor cell morphology were examined by fluorescent and transmission electron microscopy. The results of the MTT assay showed that the mucus and α-subunit of hemocyanin from the snail *H. aspersa* had the most significant antiproliferative activity of the tested samples. Cytomorphological analysis revealed that the observed antitumor effects were associated with induction of apoptosis in the tumor cells. The presented data indicate that hemocyanins and mucus from *H. aspersa* have an antineoplastic activity and potential for development of novel therapeutics for treatment of colorectal carcinoma.

## 1. Introduction

Neoplastic diseases are characterized as having high incidence and mortality and have significant health and social impacts. Conventional anticancer treatment includes a combination of surgery, radiation therapy, and chemotherapy [1]. However, traditional therapy has several drawbacks such as multidrug resistance, low selectivity, and toxicity to healthy tissues associated with severe side effects. Finding of selective and more efficient new drugs is one of the greatest challenges for pharmacology and medicine. Today, an extensive research effort has been focused on screening of compounds of natural origin with higher specificity and less adverse side effects [2]. In this context, bioactive compounds isolated from mollusk species attract a significant interest as good drug candidates for cancer therapeutic applications [3].

Numerous studies have indicated that hemocyanins, oxygen-carrying hemolymph metalloproteins, have a significant immunostimulatory and anticancer activity [4,5,6,7]. Molluscan hemocyanins are glycoproteins with high molecular masses and complex quaternary and oligosaccharide structures. They are usually composed of several structural subunits with approximate masses of 350–450 kDa, each consisting of seven or eight globular functional units connected by linker peptide strands, forming hollow cylinders [8]. The primary amino acid sequences of molluscan hemocyanins are highly divergent from mammalian sequences, which results in strong activation of the immune system. In addition, the carbohydrate moieties present in molluscan hemocyanins are considered responsible for their high immunogenicity. The carbohydrate structures of hemocyanins have been extensively studied in order to understand their organization, antigenicity, and biomedical properties [9,10]. The carbohydrate component of hemocyanins have been reported to be up to 9% (*w*/*w*) and contain diverse sugar moieties, including mannose, d-galactose, fucose, N-acetyl-d-galactosamine, and N-acetyl-glucosamine residues, as well as xylose, which is not usually present in animal proteins [11]. Hemocyanins are characterized with the presence of numerous N-glycosylation sites and limited number of O-glycosylation sites [12]. Due to these structural properties, hemocyanins stimulate the mammalian immune system nonspecifically by interacting with macrophages, polymorphonuclears, CD4+, and CD8+ cells and induce potent humoral and cellular immune response [11,13]. Moreover, significant direct antitumor effects of hemocyanins have been established in various in vitro and in vivo tumor models [4,14,15,16].

The present study aimed to assess the in vitro antitumor activity of hemocyanins isolated from *Helix aspersa*, *Helix lucorum*, and *Rapana venosa* against colorectal carcinoma cell line HT-29 and to investigate morphological and ultrastructural alterations in the tumor cells.

## 2. Materials and Methods

### 2.1. Materials

Membranes were purchased from Millipore Ultrafiltration Membrane Filters, regenerated cellulose. 3-(4,5-Dimethylthiazol-2-yl)-2,5-diphenyltetrazolium bromide (MTT), ethidium bromide (EB), and acridine orange (AO) were purchased from Sigma-Aldrich, Schnelldorf, Germany. All culture reagents, Dulbecco’s modified Eagle’s medium (DMEM; Sigma-Aldrich, Schnelldorf, Germany), fetal bovine serum (FBS; Gibso/BRL, Grand Island, NY), L-glutamine, penicillin, and streptomycin (LONZA, Cologne, Germany) were used as received. The disposable consumables were supplied by Orange Scientific, Braine-l’Alleud, Belgium. HT-29 cell line—human colorectal adenocarcinoma was obtained from American Type Cultures Collection (ATCC, Rockville, MD, USA).

### 2.2. Isolation of the Hemocyanin and Isoforms from Snail Rapana Venosa

The hemolymph was collected from *R. venosa* marine snails living in the Black Sea after cutting the foot muscles, and it was filtrated and centrifuged at 10,000 rpm and 4 °C for 20 min to remove rough particles and haemocytesas [17]. The crude hemolymph extract was ultrafiltrated (using membrane 100 kDa Millipore Ultrafiltration Membrane Filters) and the fraction above 100 kDa containing predominantly native RvH was ultracentrifugatied at 22,000 rpm and 4 °C for 180 min with rotor Kontron-Hermle A8.24 (centrifuge CENTRIKON). The sediment containing the native RvH was solubilized at a concentration of about 10% in 50 mM Tris buffer (pH 7.5) and was purified by gel filtration on column Sephadex G-200. Dissociation of native RvH was achieved by dialyzing the protein against 0.13 M glycine/NaOH buffer, pH 9.6. The structural subunits RvH1 and RvH2 were separated on an ion-exchange chromatography by a 16/10 Q Sepharose High Performance column equilibrated with 50 mM Tris/HCl buffer and 10 mM EDTA (pH 8.5) with a linear gradient of 0.0–0.5 M NaCl by FPLC system.

### 2.3. Isolation of the Native Hemocyanin and Isoforms from Snail H. Lucorum

The hemolymph was collected from the foot of garden snail *H. lucorum* and rough particles and hemocytes were removed after filtration and centrifuged at 10,000 rpm and 4 °C for 20 min [18]. After ultrafiltration of supernatant by membrane of 100 kDa (Millipore Ultrafiltration Membrane Filters, Regenerated cellulose), we applied the fraction with molecular mass above 100 kDa, containing mostly hemocyanin, to ultracentrifugation at 22,000 rpm and 4 °C for 180 min with rotor Kontron-Hermle A8.24 (centrifuge CENTRIKON). The sediment with total hemocyanin was solubilized in concentration of about 5% in 0.1 M sodium acetate buffer, pH 5.7.

The isoform βc-HlH was isolated after precipitation of native HlH during 4–5 days dialysis against 10 mM sodium acetate buffer (pH 5.1) at 4 °C and the buffer was renewed every 12 h. The βc-HlH was sedimented by centrifugation at 15,000 × *g*, at 4 °C for 30 min. The precipitate was dissolved in 100 mM sodium phosphate buffer (pH 6.5) and further purified by anion exchange chromatography on a 16/10 Q Sepharose High Performance column using a linear NaCl gradient (0.0–0.5 M) in 50 mM Tris–HCl buffer, pH 7.8.

Both subunits, *α_D_*-HlH and *α_N_*-HlH, dissolved in the supernatant, were purified by gel filtration chromatography on a Sephacryl S 300 column, were equilibrated and eluted with 50 mM Tris buffer (pH 7.5), and further concentrated by ultrafiltration (100 kDa, Millipore Ultrafiltration Membrane Filters, regenerated cellulose).

### 2.4. Isolation of the Native Hemocyanin and Mucus from Snail H. Aspersa

The native HaH was isolated after concenrtation of the hemolymph collected from the foot of garden snail *H. aspersa* by ultrafiltration (using 100 kDa, Amicon PM membranes) [4]. After ultracentrifugation at 22,000 rpm (rotor Kontron-Hermle A8.24, centrifuge CENTRIKON) and 4 °C for 3 h, the native HaH was sedimented. After removal of the supernatant, the precipitated HaH was solubilized in 50 mM Tris buffer (pH 7.5) containing 20 mM CaCl_2_ and 10 mM MgCl_2_ and further purified by gel filtration chromatography on a Sepharose 6B column (90 × 2.4 cm).

### 2.5. Separation of H. Aspersa Hemocyanin Isoforms

Three isoforms (α_D_-HaH, α_N_-HaH, and βc-HaH) were separated after 4 days of dialysis against 10 mM sodium acetate buffer (pH 5.3) at 4 °C, renewed every 12 h. The isoform βc-HaH precipitated after centrifugation and solubilized in 0.1 M sodium phosphate buffer (pH 6.5), and was further purified by gel filtration chromatography on a Sepharose 6B column and eluted with buffer of 50 mM Tris-HCl, pH 7.5. Two α-isoforms (α_D+N_) in supernatant were concentrated by ultrafiltration (Millipore Ultrafiltration Membrane Filters, regenerated cellulose) and purified of FPLC-system by a 16/10 Q Sepharose High Performance column using a linear NaCl gradient (0.0–1.0 M) in 50 mM Tris–HCl buffer, pH 8.2.

The mucus was collected from the foot of *H. aspersa* snails that were grown in Bulgarian eco-farms. After several steps of purification and homogenization, including filtration and centrifugation for removal of rough particles, the crude mucus extract was obtained [19].

Mucus extract from *H. aspersa* was analyzed by sodium dodecyl sulphate polyacrylamide gel electrophoresis (SDS-PAGE) with the molecular weight marker ranging from 250 kDa to 10 kDa using a 5% stacking gel and 12% resolving gel, according to Laemmli method with modifications [20]. All tested hemocyanins and their isoforms were analyzed by 8% polyacrylamide gel electrophoresis under native conditions, as described [21].

### 2.6. Cell culture and Cell Viability

HT-29 and Balb/c 3T3 cells were cultured in 75 cm^2^ tissue culture flasks in Dulbecco’s modified Eagle’s medium supplemented with 10% fetal calf serum, 2 mM glutamine, and the antibiotics penicillin (100 U mL^−1^) and streptomycin (100 µg mL^−1^) at 37 °C and 5% CO_2_ and 90% relative humidity.

Cell viability was assessed by MTT (methyl thiazol tetrazolium bromide) assay, as described previously [22]. Briefly, cells were plated in a 96-well microtiter plate at a density of 1 × 10^4^ cells per well in a final volume of 100 μL DMEM medium. HT-29 cells were treated with six different concentrations of the hemocyanin samples (31.25–1000 μg/mL) for 72 h. Parallel experiments with the same treatment and incubation regimen were performed on Balb/c 3T3 cells to assess the cytotoxic activity of the tested samples in non-tumor cells. After treatment, the cells were incubated with MTT dye at a concentration of 50 μg/100 μL for 3 h at 37 °C. The cells were thereafter lysed with DMSO/96% ethanol (1:1 *v*/*v*) solution. Absorbance of the reduced intracellular formazon product was read at 570 nm in a microtiter plate reader (TECAN, Sunrise, Groedig/Salzburg, Austria).

### 2.7. Fluorescent Microscopy

The morphological alterations in HT-29 cells induced by the test samples that showed higher cytotoxic activity were analyzed by fluorescent microscopy. HT-29 cells were cultured on 13 mm diameter cover glasses in 24-well plates and were treated for 24 h with hemocyanins in concentrations approximating the IC_50_ value established by the MTT test and concentrations lower and higher than the IC_50_ value. Cells treated with the standard anticancer drug doxorubicin were used as a positive control for the experiments. Doxorubicine was applied at concentrations equal to the IC_50_ value (2.7 µg/mL) established in our previous studies. The control and treated cells were stained by two different methods.

#### 2.7.1. Acridine Orange/Ethidium Bromide Double Staining

Acridine orange (AO) and ethidium bromide (EB) (live/dead) staining was performed as previously described [23]. Briefly, cell preparations of HT-29 cells were stained with the fluorescent dyes AO (5 µg/mL) and EB (5 μg/mL) in phosphate-buffered saline (PBS) and mounted on microscope slides.

#### 2.7.2. DAPI Staining

The alterations in the nuclear morphology of the tumor cells induced by hemocyanins were studied after staining with DNA binding dye 4′,6-diamidine-2′-phenylindole dihydrochloride (DAPI). The cells were fixed with methanol, incubated for 15 min in 1 µg/mL DAPI in methanol in the dark, and mounted with glycerol on microscope slides.

Stained cells were visualized and examined under a fluorescence microscope (Leica DM 5000B, Wetzlar, Germany).

### 2.8. Transmission Electron Microscopy (TEM)

Ultrastructural studies of cells exposed to bioactive compounds isolated from *Helix aspersa* at cytotoxic doses (concentrations equal or close to their IC_50_ values) were processed according to routine techniques for this type of assay. They were fixed for 1 h with 2.5% glutaraldehyde in 0.1 M pH 7.3 phosphate buffer, postfixed for 2 h in 1% OsO_4_, dehydrated, and included in Durcupan ACM Fluka. Ultra-thin sections were prepared on Reichert Ultramicrotome and stained with 2% uranyl acetate and 2% lead citrate. For the TEM study, we used an Opton transmission electron microscope.

### 2.9. Statistical Analysis

Statistical analysis was performed by one-way ANOVA followed by Bonferroni’s post hoc test (GraphPad Prism software package). *p* < 0.05 was accepted as the lowest level of statistical significance. Nonlinear regression (curve fit) analysis (GraphPad Prism) was applied to determine the concentrations inducing 50% inhibition of the cell growth (IC_50_ values).

## 3. Results

### 3.1. Isolation of Bioactive Compounds from the Three Mollusk Species

Total hemocyanins from garden snails *Helix aspersa* and *Helix lucorum* (HaH-total; HlH-total), their isoforms (subunits βc-HaH, α-HaH, βc-HlH, and α-HlH), *Helix aspersa* mucus, and subunits of *Rapana venosa* hemocyanin (RvH I and RvH II) were isolated and purified as previously described [4,17,18]. Hemocyanins are freely dissolved in the hemolymph of species in *Mollusca* and Arthropoda as a major protein constituent (90–98%) of this fluid. Specific absorption coefficient A_278_ nm = 1.413 mL·mg^−1^·cm^−1^ for HaH was used for determination of the protein concentration [24].

The hemolymph was collected from *R. venosa* marine snails living in the Black Sea and garden snails *H. aspersa* and *H. lucorum* after cutting the foot muscles [17,18,19]. The hemocyanins with molecular mass ≈ 8 MDa were isolated from the crude hemolymph extract after ultrafiltration using membrane 100 kDa. Native RvH was obtained after ultracentrifugation at 22,000 rpm and 4 °C for 180 min and purified by gel filtration on column Sephadex G-200. After dissociation of native RvH by dialyzing the protein against 0.13 M glycine/NaOH buffer (pH 9.6) two structural subunits RvH1 and RvH2 were separated on an ion-exchange chromatography by a 16/10 Q Sepharose High Performance column equilibrated with 50 mM Tris/HCl buffer and 10 mM EDTA (pH 8.5) with a linear gradient of 0.0–0.5 M NaCl by FPLC system [17].

The native *H. lucorum* [18] and *H. aspersa* [4] hemocyanins are organized by three structural subunits (βc-HaH, α_D_-HaH, and α_N_-HaH) with molecular weight (MW) ≈ 450 kDa. The βc-isoforms were precipitated from the hemolymph and further purified by gel filtration chromatography on a Sepharose 6B column, and eluted with buffer 50 mM Tris-HCl, pH 7.5. After removal of βc-isoform, both α-isoforms in supernatant were purified of FPLC-system by gel filtration chromatography on a Sephacryl S 300 column, equilibrated and eluted with 50 mM Tris buffer, pH 7.5

The mucus collected from the foot of *H. aspersa* snails was purified after filtration and centrifugation of the crude mucus extract [19].

The tested hemocyanins were analyzed by 8% native PAGE (Figure 1) [21] to confirm their molecular masses and purity. As shown in Figure 1a, purity of total hemocyanin hemocyanins (line 2 of *H. aspersa*, line 5 of *R. venosa*, and line 8 of *H. lucorum*) and their structural subunits (lines 3, 4, 6, 7, and 10) is about 90%. Line 9 shows two main bands that correspond to the two isoforms αN-HlH and αD-HlH.

The 12% SDS-PAGE analysis of the mucus extract showed that the mucus is a complex mixture of various biological substances such as antimicrobial peptides and proteins (Figure 1b, line 2).

### 3.2. Effects of the Isolated Bioactive Compounds on the Viability and Proliferative Activity of HT-29 Tumor Cells

The effects of the hemocyanin samples on the viability and proliferative activity of the colorectal adenocarcinoma cells were assessed by MTT assay after 72 h of treatment (Figure 2).

The results of the MTT assay showed that the total hemocyanins of *H. aspersa* and *H. lucorum* did not significantly affect the viability of the colon carcinoma cells. However, the isolated structural subunits α-HaH, βc-HlH, and α-HlH of hemocyanins as well the mucus from snail *H. aspersa* induced significant (*p* < 0.001 as compared to the untreated control) and dose-dependent reduction of the cell viability and proliferation.

The βc-HaH subunit induced the highest inhibition of cell viability at a concentration of 1000 µg/mL compared to the other tested subunits. Both subunits, RvH I and RvH II, of *R. venosa* hemocyanin significantly reduced (*p* < 0.001) the cell viability only at higher concentrations (500 and 1000 µg/mL, respectively). Of all tested samples, the subunits α-HaH and the mucus of snail *H. aspersa* showed the highest in vitro antitumor activity against HT-29 colon carcinoma cells. On the basis of the results of the MTT assay, we calculated the half-maximal inhibitory concentrations (IC_50_) and compared them to the IC_50_ determined by MTT assay on Balb/c3T3 cells (Table 1).

The results showed that HT-29 cells were more sensitive to the aniproliferative and cytotoxic effects of the tested samples than Balb/c3T3 cells.

### 3.3. Apotogenic Effects of the Isolated Bioactive Compounds

#### 3.3.1. Vital Double Staining of HT-29 Tumor Cells with Acridine Orange/Ethidium Bromide Acridin

For assessment of the apoptogenic potential of the selected samples, we examined the morphological alterations in HT-29 tumor cells. For this purpose, tumor cells were treated with three different concentrations of subunit α-HaH and *H. aspersa*. The changes in the cell morphology induced by the hemocyanins were examined under fluorescent microscope after staining with AO/EB (Figure 3).

Control cells were uniformly stained green and showed normal morphology and monolayer growth characteristics of the tumor cell line (Figure 3a). The positive control substance doxorubicin induced clearly pronounced apoptotic changes in the tumor cells (Figure 3b). Distinct dose-dependent morphological changes were found in HT-29 cells treated with α-HaH and the mucus of *H. aspersa*. The cell density of the monolayer was reduced and cells with intensive green fluorescence indicative of early apoptotic chromatin condensation changes were observed in treated cell cultures. In addition, late apoptotic cells with condensed chromatin, fragmented nuclei, and red-orange staining indicating the loss of membrane integrity and entry of ethidium bromide into the cell were also present.

#### 3.3.2. DAPI Staining of HT-29 Tumor Cells

Further, the alterations in the nuclear morphology of the HT-29 cells induced by subunits α-HaH and the mucus were studied by fluorescent microscopy after staining with DNA-binding dye DAPI (Figure 4).

The control HT-29 tumor cells showed typical morphology of the nucleus with homogenous blue staining, and cells in mitosis phase were observed. The nuclei of the cells treated with the α-HaH subunits and the mucus were irregular in shape, more brightly colored, and had intense condensation of chromatin. Some treated cells showed nuclear fragmentation and formation of apoptotic bodies. Mitotic figures were not found in cells exposed to hemocyanin and mucus.

### 3.4. Transmission Electron Microscopy

The ultrastructural alterations induced by the subunits α-HaH and the mucus from garden snail *H. aspersa* in HT-29 tumor cells were examined by transmission electron microscopy (Figure 5).

Our TEM observations of HT-29 cells in the control culture showed bipolar elongated cells with normal morphology (Figure 5a). The nucleus and the cytoplasm of the untreated cells had a normal structure, were relatively electronically dense, and were without cellular inclusions. The nucleus was more centrally located and several vesicles and the endoplasmic reticulum was peripherally observed. The cells’ nuclei exhibited adenocarcinoma-specific appearance with prominent nucleoli. The notable superficial cellular membrane protrusions/microvilli evenly distributed on the surface of the plasmalemma are also typical for the carcinoma cells.

The ultrastructural aspects of the HT-29 cells treated with different test compounds showed mild to more pronounced changes (Figure 5b). The subunits α-HaH and the mucus extract seriously affected cell morphology. The lesions were mainly expressed in loss of polarity and rounded shapes. Numerous cytoplasmic vesicles were observed, being dispersed throughout the cytoplasm along with single electron-dense objects of a larger number and size (likely disorganization of the cytoskeleton and vacuolization of the organelles along the apoptosis pathway). No nuclear fragmentation was found after subunits’ α-HaH exposure. The observed cells were also characterized by enlightened nuclei and cytoplasm, budding to form apoptotic bodies including parts of the cytoplasm.

The cells treated with the *Helix aspersa* mucus showed changes similar to the enlightened nuclei presented above, with separate small condensates of heterochromatin, focal perinuclear expansions of the membrane space of endoplasmic reticulum, and abundant organelle vacuolization (Figure 5c). The appearance of different electron-dense bodies and single autophagosomes or mitophagosome-like structures in cells were observed. Rare nuclear fragmentation, formation of apoptotic bodies with parts of the cytoplasm, and vacuoles within them were found as serious morphological alterations. In addition, extended nuclei were observed here, leading to changed nucleus–cytoplasm index.

## 4. Discussion

The search for novel, more effective, and safe antitumor medicines with natural origin is one of the main trends of contemporary oncology research. In this respect, hemocyanin oligomeric copper-containing glycoproteins that function as oxygen carriers in the hemolymph of mollusk and arthropod species represent significant interest because they combine strong immunostimulating activity and direct anticancer effects [5,6,25,26,27]. Among this class of compounds, the hemocyanin obtained from *Megathura crenulata*, known as keyhole limpet hemocyanin (KLH), has been most extensively studied and has found a number of biotechnological and medical applications. Clinical studies have shown that KLH treatment significantly reduces the tumor recurrence of patients with urinary bladder carcinoma [11]. It has also found an application as a bio-adjuvant and protein carrier in experimental antiviral and anticancer vaccines [16,28,29]. Moreover, KLH has been reported to induce significant reduction in the proliferation and viability of prostate cancer cells, estrogen-dependent breast and estrogen-independent breast cancer cells, and Barrett’s esophageal adenocarcinoma cells [14,30].

The diverse biological activities and increasing biomedical applications of KLH have led to growing interest and a search of other hemocyanins with similar or more potent immunostimulatory and antitumor properties. In the present study, the antitumor activity of total hemocyanins isolated from *H. aspersa*, *H. lucorum,* and their subunits; subunits of *R. venosa* hemocyanin; and *H. aspersa* mucus was examined in the HT-29 human colorectal carcinoma cell line. The results showed that the total hemocyanins did not significantly affect the viability of the colorectal carcinoma cells, while the isolated hemocyanin subunits induced a statistically significant decrease of the tumor cell growth. Similarly, in a previous study, it was found that the subunits of *H. lucorum* hemocyanin induce stronger inhibition of the tumor growth of bladder carcinoma CAL-29 cells as compared to the effect of native HlH [26]. It could be supposed that the potent tumor inhibiting activity is due to the specific oligosaccharide structures, which are more easily accessible in the isolated structural subunits. In addition to hemocyanin subunits, the *H. aspersa* mucus also showed a significant anticancer effect against HT-29 carcinoma cells. The bioactive compounds, structural subunits α-HaH, βc-HaH, and mucus, isolated from garden snail *H. aspersa,* appeared to be the most active of all tested samples in inhibiting the colon cancer cell growth. This result is in agreement with the data reported by Matusiewicz et al., indicating that the application of extracts from lyophilized mucus and foot tissues of *H. aspersa* decrease the viability of the colon cancer cell line Caco-2 [31].

The reduction of the viability the HT-29 cells induced by treatment with HlH subunits was statistically significant with a clear dose dependency, but it was slightly weaker than those of HaH subunits. The subunits of *R. venosa* hemocyanin showed an antitumor effect only at the higher tested concentrations. Hemocyanins isolated from the land snail *H. pomatia* and marine snail *R. thomasiana* were previously found to express strong immunostimulatroy action and to inhibit tumor cell growth in a murine model of colon carcinoma [32]. These findings taken together with the results of the present study demonstrate that the hemocyanins isolated from different mollusk species have significant antitumor effects against colorectal carcinoma.

The tested bioactive substances, mucus and α-HaH from snail *H. aspersa,* which showed higher antiproliferative activity in the MTT assay, were further used in morphological studies that aimed to analyze the mechanisms that mediate their anticancer action and the nature of the cell death induced in HT-29 carcinoma cells. The fluorescent and transmission electron microscopy studies revealed typical apoptotic alterations in the cellular and nuclear morphology of the tumor cells treated with the tested samples. Apoptotic cell death is an important biological mechanism that contributes to the maintenance and integrity of multicellular organisms and an important factor in preventing cancer. Thus, the ability to induce apoptosis in tumor cells is a desired property of the anticancer therapeutics.

Our results are in line with the previously published data, indicating a significant proapoptotic activity of molluscan hemocyanins in various tumor cell lines [4,33,34,35] and suggesting their potential use in anticancer therapy.

## 5. Conclusions

The tested hemocyanin samples isolated from garden snails *H. aspersa* and *H. lucorum*, marine snail *R. venosa*, as well as the mucus from garden snail *H. aspersa* significantly decreased the cell viability of HT-29 carcinoma cells. The mucus and α-HaH from snail *H. aspersa* were identified as bioactive substances with higher antiproliferative activity against HT-29 carcinoma cells. The mechanism of their antitumor activity includes the induction of apoptosis. In the combination with their already known immunogenic effect, these findings support further studies of molluscan hemocyanins as potential therapeutic agents against colorectal cancer.

## Figures and Tables

**Figure 1 biomedicines-08-00194-f001:**
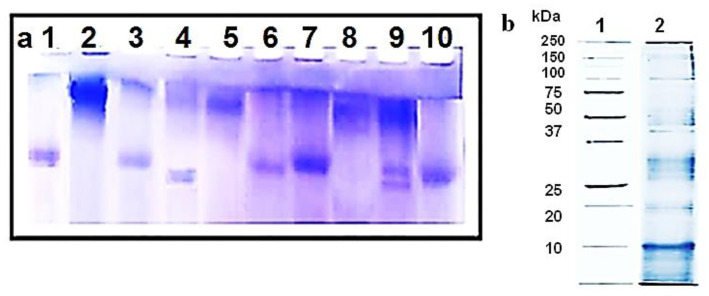
(**a**) The 8% native gel electrophoresis of the tested hemocyanins with Coomassie Blue G-250 dye: positions (1) standard ferritin (450 kDa); (2) total *Helix aspersa* hemocyanin; (3) two α-isoforms of *H. aspersa* hemocyanin; (4) structural subunit βc-HaH; (5) total *Rapana venosa* hemocyanin; (6) structural subunit RvH I; (7) structural subunit RvH II; (8) total *Helix lucorum* hemocyanin; (9) two α-isoforms of *H. lucorum* hemocyanin; (10) structural subunit βc-HlH. (**b**) The 12.0% SDS-PAGE analysis visualized by staining with Coomassie Blue G-250: (1) molecular weights of standard proteins from Bio-rad; (2) mucus extract from *H. aspersa*.

**Figure 2 biomedicines-08-00194-f002:**
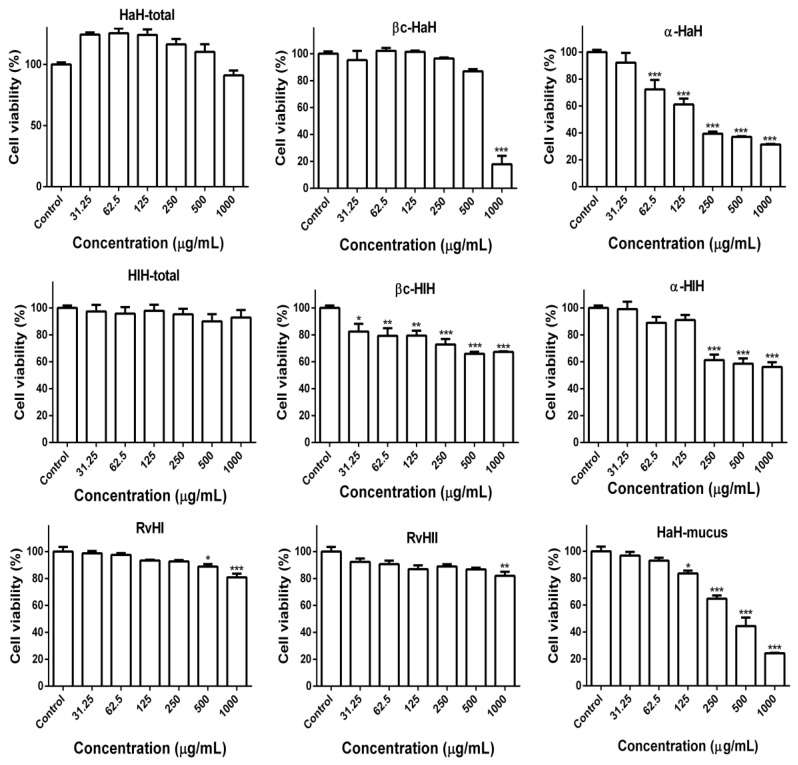
Antiproliferative effect of hemocyanins from *H. aspersa* (HaH-total) and its structural subunits βc-HaH and α-HaH; hemocyanin from *H. lucorum* (HlH-total) and its subunits βc-HlH and α-HlH; subunits RvH I and RvH II of hemocyanin from *R. venosa* (RvH); mucus extract of *Helix aspersa* on HT-29 colorectal carcinoma cell line. * *p* < 0.05; ** *p* < 0.01; *** *p* < 0.001

**Figure 3 biomedicines-08-00194-f003:**
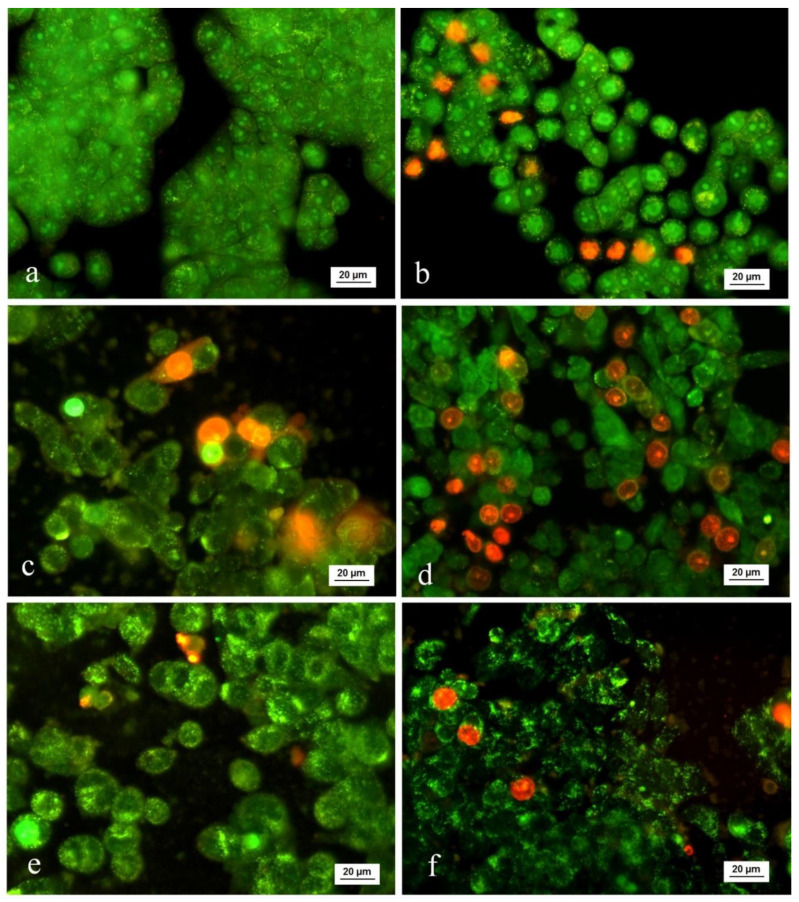
Fluorescence microscopic images of acridine orange (AO)/ethidium bromide (EB)-stained HT-29 colorectal carcinoma cells after treatment with bioactive compounds isolated from *H. aspersa*: (**a**) control; (**b**) doxorubicin 2.7 µg/mL; (**c**) subunit α-HaH 200 µg/mL; (**d**) subunit α-HaH 800 µg/mL; (**e**) *H. aspersa* mucus 400 µg/mL; (**f**) *H. aspersa* mucus 800 µg/mL.

**Figure 4 biomedicines-08-00194-f004:**
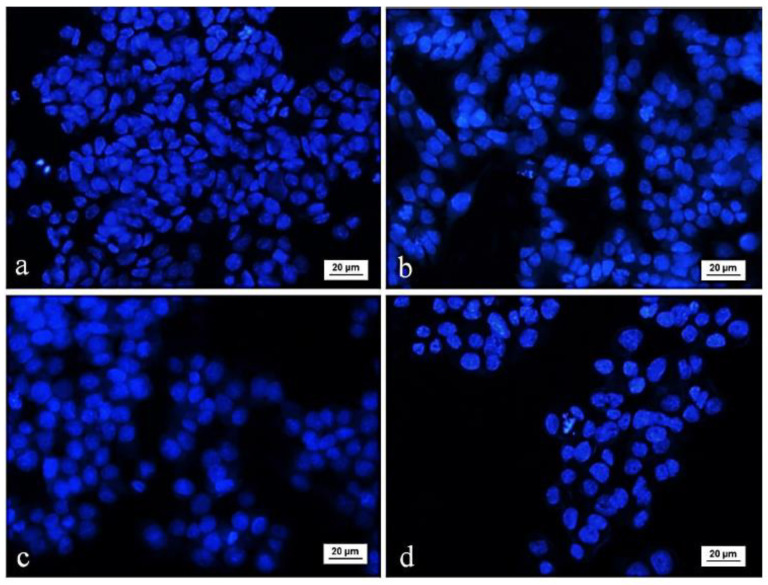
Fluorescence microscopic images of 4′,6-diamidine-2′-phenylindole dihydrochloride (DAPI)-stained HT-29 colorectal carcinoma cells after treatement with bioactive compounds isolated from garden snail *Helix aspersa*: (**a**) control; (**b**) *Helix aspersa* mucus 400 µg/mL; (**c**) subunit α-HaH 100 µg/mL; (**d**) subunit α-HaH 200 µg/mL.

**Figure 5 biomedicines-08-00194-f005:**
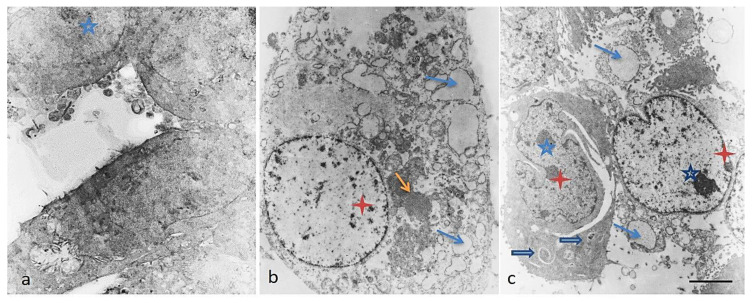
Transmission electron microscopy images of HT-29 cells after treatment with bioactive compounds isolated from *Helix aspersa*. (**a**) Control HT-29 cells with normal morphology: blue five-point star-nucleolus. (**b**) HT-29 cells treated with the subunits α-HaH with affected cell morphology: red four-point star—granules of highly condensed chromatin, thin blue arrow—numerous vacuoles, thin orange arrow—electron-dense cytoplasmic structures. (**c**) HT-29 cells treated with the mucus of *Helix aspersa* with impaired morphology: blue five-point star—nucleolus, red four-point star—granules of highly condensed chromatin, thin blue arrow—numerous vacuoles, blue filled arrow—presence of single autophagosomes or mitophagosome-like structures. TEM scale bar = 2 µm.

**Table 1 biomedicines-08-00194-t001:** The half-maximal inhibitory concentrations (IC_50_) of hemocyanins and their subunits isolated from *H. aspersa*, *H. lucorum*, and *R. venosa* determined by MTT assay on HT-29 and Balb/c3T3 cells.

IC_50_ Values (µg/mL)	HT-29	Balb/c 3T3
HaH-total	›1000	934.1
Subunit βc-HaH	733.8	›1000
Subunits α-HaH	235.3	514.6
HlH-total	›1000	›1000
Subunit βc-HlH	›1000	›1000
Subunits α-HlH	›1000	›1000
Subunit RvH I	›1000	›1000
Subunit RvH II	›1000	›1000
Ha-mucus	415.7	825.1

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
