# Peer review of "Hemocyanins from Helix and Rapana Snails Exhibit in Vitro Antitumor Effects in Human Colorectal Adenocarcinoma"

_biomedicines, 2020, doi:10.3390/biomedicines8070194_

Round 1
Reviewer 1 Report
The paper submitted for publication in “Biomedicines” by Georgieva et al. is related to the evaluation of the antitumor effects of hemocyanins isolated from Helix and Rapana snails in HT-29 human colorectal adenocarcinoma cells. Authors succeeded in showing that tested hemocyanins inhibit proliferation and might induce apoptosis of HT-29 cells.
However, there are several aspects that should be considered.
1) The authors described in detail the isolation of bioactive hemocyanins and their isoforms. However, they should show data on identification of isolated bioactive substances. The structures of the isolated compounds must be determined by using nuclear magnetic resonance (NMR) and mass spectrometry (MS).
2) In the Introduction (lanes 35-36) authors stated that “Today an extensive research effort has been focused on screening of compounds of natural origin with higher specificity and less adverse side effects.” To increase value of the manuscript the authors have to test the antiproliferative activities of isolated hemocyanins towards normal epithelial cells. For example, CCD 841 CoN normal human colonic epithelial cells might be tested and compared to HT-29 human colorectal adenocarcinoma cells.
3) Besides the AO/EB assay and DAPI staining, authors need perform an Annexin-V/PI analysis to provide statistically robust evidence on potential of alpha-HaH and Ha-mucus substances to induce apoptosis in HT-29 cells. Additionally, the markers of apoptosis (caspases, cytochrome c, Bax/Bcl-2, PARP-1, etc.) should be analyzed by western blotting.
Author Response
Thanks to the reviewers for the recommendations made to improve the presented results. Thanks to these recommendations, we have added additional results from our research, which we present:
REVIEWER 1
1) The authors described in detail the isolation of bioactive hemocyanins and their isoforms. However, they should show data on identification of isolated bioactive substances. The structures of the isolated compounds must be determined by using nuclear magnetic resonance and mass spectrometry.
Answer : Page 3, Line 120-124 in section 2. Materials and Methods
Mucus extract from H. aspersa was analysed by sodium dodecyl sulphate-polyacrylamide gel electrophoresis (SDS-PAGE) with the molecular weight marker ranging from 250 kDa to 10 kDa using a 5% stacking gel and 12% resolving gel, according to Laemmli method with modifications [20]. All tested hemicyanins and there isoforms were analysed by 8% polyacrylamide gel electrophoresis under native conditions, as described [21].
The isolated compounds are with molecular weight above 450 kDa and nuclear magnetic resonance and mass spectrometry cannot be applied.
2) In the Introduction (lanes 35-36) authors stated that “Today an extensive research effort has been focused on screening of compounds of natural origin with higher specificity and less adverse side effects.”To increase value of the manuscript the authors have to test the antiproliferative activities of isolated hemocyanins towards normal epithelial cells. For example, CCD 841 CoN normal human colonic epithelial cells might be tested and compared to HT-29 human colorectal adenocarcinoma cells.
Answer : We agree with the reviewer that the assessment of antitumor activities of novel chemical compounds should also address their selectivity and potential toxicity to normal cells. New experimental data about the antiproliferative activity of the isolated hemocyanins towards the mouse embryonal fibroblast cell line Balb/c 3T3 have been added to the revised version manuscript. This cell line is a widely used model systems for in vitro toxicological investigations.
3) Besides the AO/EB assay and DAPI staining, authors need perform an Annexin-V/PI analysis to provide statistically robust evidence on potential of alpha-HaH and Ha-mucus substances to induce apoptosis in HT-29 cells. Additionally, the markers of apoptosis (caspases, cytochrome c, Bax/Bcl-2, PARP-1, etc.) should be analyzed by western blotting.
Answer : We agree with the reviewer that a large number of detection methods and biomarkers of apoptosis are now available. The use of molecular and flow cytometric methods is particularly important in cases when the results of the classical cytomorphological analysis are inconclusive. The results of the fluorescent microscopic analysis of cells treated with the test samples revealed clearly expressed morphological signs of apoptosis. Moreover, these results were confirmed by ultrastructural investigations. The aim of our study was to screen nine hemocyanin samples for antitumor activity and to select the most active compounds for further more detailed investigations. The subunit alpha-HaH and Ha-mucus were identified as the most active compounds among the tested samples. The molecular mechanisms of the antineoplastic effects of these compounds and their in vivo antitumor activities are objectives of our ongoing studies.
Reviewer 2 Report
The manuscript by Georgieva et al described the extraction, purification and isolation of subunits of hemocyanins from Helix and Rapana snails. The in vitro cytotoxic activity and IC50s of these extracts and isolates against a colorectal cancer HT-29 cell line were measured using MTT method. Further morphological changes in cells were measured using fluorescent microscopy after various staining methods to investigate their mechanism of action such as induction of apoptosis. Finally, TEM was used directly to observe the cellular change after hemocyanin treatment. The study is sound, properly designed and carried out. The experimental results and their significance are well discussed.
However, there are some issues to be addressed before considering for publication:
- There is a lack of characterization (e.g. electrophoresis, gel filtration, mass spectrometry, immunological detection) of the isolated hemocyanins to conclude if these isolated proteins are hemocyanins, how pure they are, and whether they are new or known hemocyanins.
- The yield of hemocanins and the method of determining their concentration were not provided.
- There is a lack of positive control and concentration-dependent experiments for fluorescent staining over the treatment of the cancer cell line.
- Scale bars in Figure 2-4 are not clearly indicated.
Author Response
Thanks to the reviewers for the recommendations made to improve the presented results. Thanks to these recommendations, we have added additional results from our research, which we present:
REVIEWER 2
1) There is a lack of characterization (e.g. electrophoresis, gel filtration, mass spectrometry, immunological detection) of the isolated hemocyanins to conclude if these isolated proteins are hemocyanins, how pure they are, and whether they are new or known hemocyanins.
Answer : Page 4, line 189-210
The tested hemocyanins were analyzed by 8% native PAGE (Figure 1), [21] to confirm theirs molecular masses and purity. The SDS-PAGE analysis of the mucus extract showed that the mucus is a complex mixture of various biological substances such as antimicrobial peptides and proteins.
Figure 1. a) 8% Native gel electrophoresis of the tested hemocyanis with Coomassie Blue G-250 dye: positions 1) standard Ferritin (450 kDa); 2) total H. aspersa hemocyanin; 3) two α–isoforms of H. aspersa hemocyanin; 4) structural subunit βc-HaH; 5) total R. venosa hemocyanin; 6) structural subunit RvH I; 7) structural subunit RvH II; 8) total H. lucorum hemocyanin; 9) two α–isoforms of H. lucorum hemocyanin; 10) structural subunit βc-HlH. b) 12.0% SDS-PAGE analysis visualized by staining with Coomassie Blue G-250: 1) molecular weights of standard proteins from Bio-rad; 2) mucus extract from H. aspersa.
2) The yield of hemocanins and the method of determining their concentration were not provided.
Answer : Page 4, line 170,171 Specific absorption coefficient A278 nm = 1.413 ml·mg−1·cm−1 for HaH was used for determination of the protein concentration [24].
3) There is a lack of positive control and concentration-dependent experiments for fluorescent staining over the treatment of the cancer cell line.
Answer : Figure 2 have been replaced in the corrected manuscript. Microphotographs of cell treated with doxorubicin and cells treated with two different concentrations of α-HaH are presented at the new figure.
4) Scale bars in Figure 2-4 are not clearly indicated.
Answer : We agree with the reviewer’s comment. The figures have been corrected and added to the revised version of the manuscript.
The presented results can be improved
Answer: The presented results have been improved by adding new additional results.
- The methods described can be improved
Answer: The described methods have been improved to include additional methods of analysis, such as:
- The design of research needs to be improved
Answer: The performance of research has been improved by presenting new figures - The conclusions, supported by the results, need to be improved
Answer: The conclusions have been improved and supplemented with the new presented results
Round 2
Reviewer 1 Report
The authors have addressed satisfactorily the concerns previously raised. I have not any further objections.
Author Response
Thanks to the reviewers for the recommendations made to improve the presented results. Thanks to these recommendations, we have added additional text, and English language and style were corrected.
Reviewer 2 Report
The revision has improved and fairly addressed the concern. The yield (based on starting material as %) and purity (how pure they are as %) of each isolated hemocyanin based on gel electrophoresis image are still not clearly stated. If not pure in some case, please make comments.
Author Response
Comments and Suggestions for Authors
The revision has improved and fairly addressed the concern. The yield (based on starting material as %) and purity (how pure they are as %) of each isolated hemocyanin based on gel electrophoresis image are still not clearly stated. If not pure in some case, please make comments.
Answer :
We accept the referee's remark, and with the added text we have supplemented the information about the purity of the tested samples.
Line 168 -174 The isolation of hemocyanin and isoforms from the sea snail Rapana venosa and the garden snails H. lucorum and H. aspersais is described in ref. [17,18 and 19 respectively]. Hemocyanin is freely dissolved in the hemolymph of species in Mollusca and Arthropoda as a major protein constituent (90–98%) of this fluid. Specific absorption coefficient A278 nm = 1.413 ml·mg−1·cm−1 for HaH was used for determination of the protein concentration [24].
Line 191 -195 As is shown in Figure 1a, 8% PAGE the purity of the tested hemocyanis (line 2 - total H. aspersa hemocyanin, line 5 - total R. venosa hemocyanin and line 8 - total H. lucorum hemocyanin) and their structural subunits (lines 3,4,6, 7 and 10) is about 90%. Line 9 shows two main bands, which correspond to the two isoforms αN - HlH and αD - HlH.
Line 191 -195 The 12% SDS-PAGE analysis of the mucus extract showed that the mucus is a complex mixture of various biological substances such as antimicrobial peptides and proteins (Figure 1b, line 2).